

# SANgo: a storage infrastructure simulator with reinforcement learning support

Kenenbek Arzymatov[1,*], Andrey Sapronov[1], Vladislav Belavin[1], Leonid Gremyachikh[1], Maksim Karpov[1,*], Andrey Ustyuzhanin[1], Ivan Tchoub[2] and Artem Ikoev[2]

[1] National Research University Higher School of Economics, Moscow, Russia
[2] YADRO, Moscow, Russia
[*] These authors contributed equally to this work.

## ABSTRACT

We introduce SANgo (Storage Area Network in the Go language)—a Go-based package for simulating the behavior of modern storage infrastructure. The software is based on the discrete-event modeling paradigm and captures the structure and dynamics of high-level storage system building blocks. The flexible structure of the package allows us to create a model of a real storage system with a configurable number of components. The granularity of the simulated system can be defined depending on the replicated patterns of actual system behavior. Accurate replication enables us to reach the primary goal of our simulator—to explore the stability boundaries of real storage systems. To meet this goal, SANgo offers a variety of interfaces for easy monitoring and tuning of the simulated model. These interfaces allow us to track the number of metrics of such components as storage controllers, network connections, and hard-drives. Other interfaces allow altering the parameter values of the simulated system effectively in real-time, thus providing the possibility for training a realistic digital twin using, for example, the reinforcement learning (RL) approach. One can train an RL model to reduce discrepancies between simulated and real SAN data. The external control algorithm can adjust the simulator parameters to make the difference as small as possible. SANgo supports the standard OpenAI gym interface; thus, the software can serve as a benchmark for comparison of different learning algorithms.

## INTRODUCTION

A storage system is a critical part of any IT infrastructure. A significant effort is put into developing reliability techniques and failure protection schemes of storage area network systems (SAN), which is an example of a general concept of storage infrastructure. There are various means of achieving and increasing the system's reliability in terms of both accessibility and data preservation: replication of the system's physical components or software solutions such as cluster management and RAID technology. The requirements for reliability and scalability of the SAN system result in its complexity, and consequently, the SAN architecture becomes challenging to supervise. A dedicated computer simulation

Corresponding author
Maksim Karpov, mekarpov@hse.ru

model or a digital twin can be created to observe the behavior of the storage infrastructure virtually. Such computer model granularity depends on the simulation tasks, which may be performance optimization, failure diagnostics/prediction, or any other data-driven method for improving the storage system's functionality.

There are three qualitatively different approaches to create a SAN simulator:

1. a purely physical model with detailed hardware/software processes;
2. a pure Machine Learning (ML) model, based on data from previous SAN operations;
3. a hybrid of the above—when a simplified SAN architecture is implemented in simulation, and its parameters are adjusted by a trained Deep Learning algorithm.

The first method gives the best results in simulation accuracy and has outstanding potential for extrapolation beyond the known operation domain. It predicts the system's behavior with a configuration that might not have been implemented yet. However, this approach requires significant expertise in the field of storage architecture and a deep understanding of the software stack used on-board the SAN. It also takes enormous resources to implement the model with all the physical and logical details of the system components when trying to achieve the required simulation quality.

The pure ML model is based on the data-driven approach, meaning that the model is trained on the data collected from the existing storage system. This method, on the contrary, may not require as much knowledge of the SAN structure and its operation principles. It may produce satisfactory accuracy of the simulated parameters, but lack the vital feature of scalability—the model is bonded with the data collected for particular SAN architecture and configuration and is unable to extrapolate beyond the scope of the training parameters.

The hybrid simulation technique is a good trade-off in terms of the quality and depth of the domain expertise needed. It requires a relatively simple model of the SAN architecture, implementing only basic components with their functionalities and logical associations. The components must have effective adjustable parameters with meanings similar to the real ones, for example, the CPU clock speed or link bandwidth. These parameters must be adjusted by a reinforcement learning (RL) algorithm to improve the simulation quality. Similarly to the second approach, the hybrid approach needs real data for training.

The development of SANgo was motivated by research aimed to create a monitoring tool able to diagnose the current SAN state and predict possible failures of its components. The diagnostic algorithms were designed to analyze the time series of different parameters collected during storage system operation. Since the algorithms are based on the ML paradigm, they needed plenty of training data, which, in this particular case, could only be obtained from a simulated environment. An essential requirement was the physical consistency of the synthetic data, and with reasonable efforts, it became plausible with the hybrid approach to the simulation.

Such a coupling of the simplified SAN simulator with runtime control by an RL agent required a dedicated study. With many deep learning technologies available, one has to select an approach and its configuration to obtain the best combination of simulation quality, training effort, and speed. The corresponding study was conducted and presented in *Karpov et al. (2018)* and *Sapronov et al. (2018)*.

The simulator is developed to provide a user with the simplicity of configuration. The user needs to specify the following files: a file that sets the architecture of a storage array in an XML format, a file that defines the impact of external effects (temperature, humidity, atmospheric pressure, and vibration) and finally a set of functions that defines the behavioral logic of each component of a storage array.

There are several open-source storage system simulators available, for example, a software package "CODES project" (*Cope et al., 2011*; *Mubarak et al., 2017*) developed by a team of researchers from the Computer Science and Math department of Argonne National Laboratory and Rensselaer Polytechnic Institute (US, Illinois). The CODES simulator is based on the technologies of the Rensselaer's Optimistic Simulation System (ROSS), which allows the parallel execution of an event-driven system that can significantly decrease the runtime of the simulation. The main uses of CODES include large-scale storage systems, scientific distributed applications, parallel and high-performance computing systems with high-load input/output operations, and computational complexity. Another simulator, the C++ based ns3 framework (*Riley & Henderson, 2010*), is also popular among researchers. The general approach to network-like structure simulation is the OMNeT++ framework (*Varga & Hornig, 2008*). Another work presents a simulation compliant with the fiber channel technology often used in contemporary SAN architectures, developed as the SANSim tool (*Wang et al., 2003*). More simulation method descriptions and studies dedicated to SAN system modeling can be found in *Molero et al. (2000a)*, *Molero et al. (2001)*, *Perles et al. (2001)*, *Molero et al. (2000b)* and *Muknahallipatna et al. (2010)*.

## SOFTWARE DESCRIPTION

SANgo is a modular framework for the Discrete Event Simulation (DES) of storage infrastructure. Its metadata description is given in Table 1. DES is a method of simulating the behavior and performance of a real-life process, facility, or system. DES models the system as a series of events (e.g., a beginning/end of file writing, data block transfer or a start of TCP connection), which happens over time. The main assumption of the DES paradigm is the invariability and consistency in the modeled system between the events. It proves itself as a viable approach to effectively evaluating diverse sets of algorithms. More information can be found in *Fishman (1978)*.

SANgo provides the functionality to build a simple model for specific computing environments, especially storage area networks. The primary site of interest is the exploration of the behavior of the storage machine under stress testing or exploitation in the medium- or long-term, for observing failures of its components.

In SANgo, each file to be written to the storage is modeled as an independent entity that has corresponding attributes, such as name, size, and block size. The fundamental principle of the simulation process is resource modeling and control. SANgo focuses on algorithms that have the following common objective: assigning a set of tasks to a set of resources in a way that is optimal with respect to some metric. The three basic types of resources are provided: CPU, network interface, and storage. Other types are built upon these basic ones. For example, a storage controller, PCIe-fabric can use CPU type as its

**Table 1 Code metadata.**

| Nr. | Code metadata | Description |
|---|---|---|
| C1 | Current code version | v 0.8 |
| C2 | Permanent link to code/repository used for this code version | https://github.com/HSE-LAMBDA/sango |
| C3 | Code Ocean compute capsule | https://codeocean.com/capsule/9707185 |
| C4 | Legal Code License | GPL |
| C5 | Code versioning system used | git |
| C6 | Software code languages, tools, and services used | Golang 1.8+, Python 3.6, Jupyter Notebook |
| C7 | Compilation requirements, operating environments & dependencies | pytorch 0.4.0+, OpenAI/gym, libzeromq-dev |
| C8 | If available, link to developer documentation/manual | |
| C9 | Support email for questions | karzymatov@hse.ru |

basis; RAM, SSD, hard-drives, and JBOD (as a collection of hard-drives) use storage type to implement additional functionality. An example of SAN architecture with these primary resources collected into SAN components is shown in Fig. 1.

The SANgo core library is written in the Go programming language. This choice was made because Go can efficiently create lightweight threads, so-called goroutines. We use a goroutine as a representation of the SAN component logic that a user wants to simulate. In other words, the user creates a function and specifies a sequence of actions and/or behaviors that a resource should follow. Such functions are given in Table 2. When sequences of 'behaviors' are specified, the simulation starts.

## Simulation sequence

The general algorithm of the SANgo operation is shown in Fig. 2. The entry point is the definition of the input parameters: the storage system topology and component parameters and the definition of the load scenario. The latter means the sequence of I/O requests from a virtual client that the simulated SAN must process. Depending on the load scenario, a specific process (component activity) and mode (asynchronous or synchronous) are chosen. Further, an option of component failure, either spontaneous or planned, is processed. With or without this failure, the process of event simulation continues until the load scenario is complete. During each simulation stage, metrics are collected from the components and written out. These metrics describe the state of the simulated system and provide primary information on its operation. Unlike other simulators, the framework allows online adjustments of the components parameters. This feature makes possible coupling with an RL agent as described in Section "Illustrative Examples".

## Software functionalities

**Online configuration** The simulation models created within the SANgo framework are configurable during runtime. It is an important feature, created intentionally, for coupling with a controlling agent and implementing the hybrid simulation method. The details of such cooperation can be found in Section "Illustrative Examples".

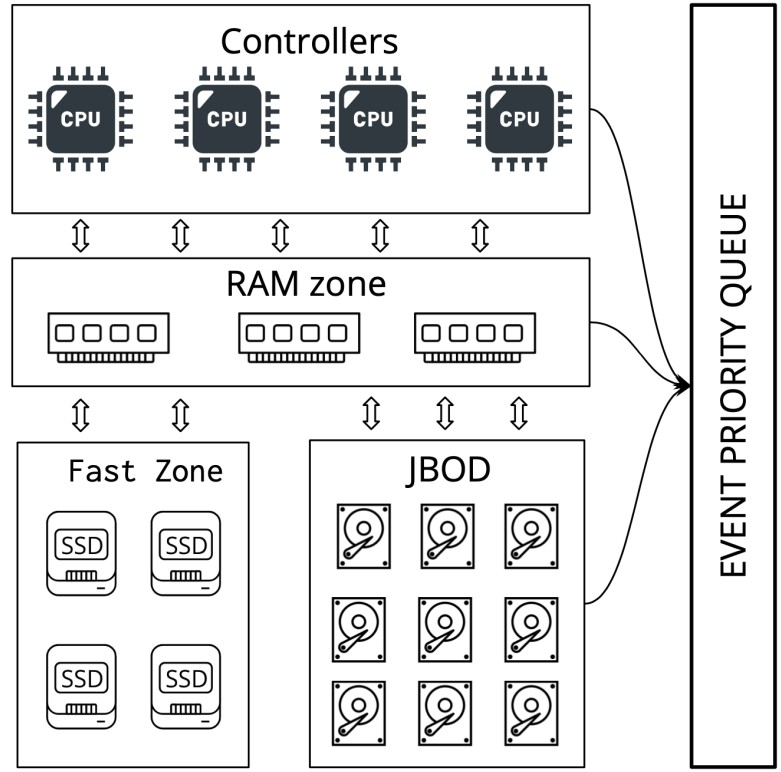

**Figure 1** An example of SAN architecture implemented in the SANgo framework.

**Table 2** Functions that define the simulated behaviors of SAN components.

| Component name | Behavior | Description |
| --- | --- | --- |
| | SendPacketSync | Send a packet to the dest and wait until the end of transmission (two side communication) |
| | SendPacketAsync | Send a packet without waiting for the end of the transmission (two side communication) |
| | BroadCastSendPacket | Send a broadcasting message with a packet to all processes which listen to 'dest' address |
| Contoller | DetachedSendPacket | Send a packet and wait until the end of the transmission (one side communication) |
| | ExecutePacket | Process packet |
| | Wait | Turn on waiting mode |
| | WriteAsync | Write data blocks in a logical volume in an async/sync mode |
| | WriteSync | |
| Hard-drive | ReadAsync | Read data blocks from a logical volume in an async/sync mode |
| | ReadSync | |

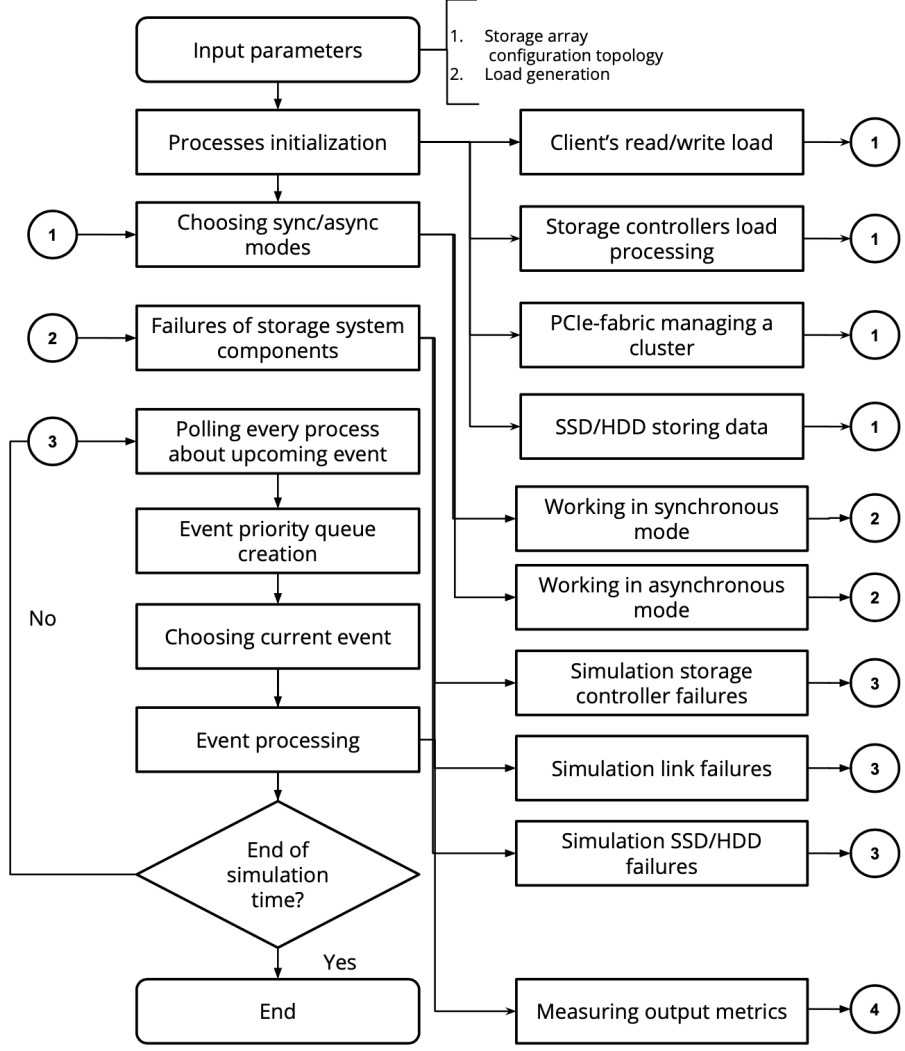

**Figure 2** SANgo discrete event simulation workflow.

**Load generation** One of the main interests is how a storage array behaves under different loads. The complete list of the fully measured output metrics is shown in Table 3.

**Output format**. During the runtime of the simulation, the program opens a file descriptor and saves the simulated metrics in a JSON (*Crockford, 2018*) array format. Each item in the array is a key-value representation of the resource objects (storage controllers, links, and hard-drives) created in the simulation. Go allows users to change the encoding of each object by the format string stored under the "json" key in the resource field's tag. This language feature allows the user to additional mute parameters. Together with native struct embedding (inheritance), it is possible to adapt the output format needed for different hardware configurations easily.

**Anomalies**. For the simplified simulation, we chose to split the failure factors into two categories: internal and external. The former relates to the component operation mode,

**Table 3  Output characteristics.**

| Parameters | Component name | Output name | Description |
|---|---|---|---|
| Inner | Storage controller | cpu_usr | User CPU utilization, % |
| | | cpu_sys | System CPU utilization, % |
| | | cpu_idle | CPU idle time, % |
| | | mem_used_MB | Used RAM memory, MB |
| | | mem_free_MB | Free RAM memory, MB |
| | | cpu_temp | CPU temperature, °C |
| | | fan_speed | Fan angular velocity, Hz |
| | Network interface | RxKB | Input traffic through interface, MB |
| | | TxKB | Output traffic through interface, MB |
| | Hard drive | tota_cap_MB | Raw capacity, MB |
| | | used_cap_MB | Used capacity, MB |
| | | alloc_MB | Memory used by system, MB |
| | | len_MB | Requested volume size, MB |
| | | r_KBps | Disk read/write speed, KB/s |
| | | w_KBps | |
| | | r_ops | Read/write operations per second, IOPS |
| | | w_ops | |
| | | r_await_ms | Average read/write request |
| | | w_await_ms | processing time, ms |
| | | ccrm | Number of denied requests, n |
| | | rrqm | Number of read/write requests, n |
| | | wrqm | |
| | | avgrq_sz | Average size of request, n |
| | | avgqu_sz | Average client queue size, n |
| | | MB_read | Total amount of read/write data, MB |
| | | MB_wrtn | |
| | | blk_read | Number of read/write blocks |
| | | blk_wrtn | with given size, n |
| | Each component | Health state | One of OK, MISSED, BAD or LOST |
| Outer | Atmospheric | humidity | Current values of humidity, atmosphere |
| | | atm_pressure | pressure, temperature, and vibration |
| | | temperature | meausured inside storage system %, kPa, |
| | | vibration | °C, Hz respectively |

its load, and neighboring component conditions. The internal factors are better described by the RL counterpart of the SANgo, which takes into account operation history and composite state of the system. For the latter category, we consider four ways in which the external parameters could affect the failure of parts of the system, such as the impact of temperature, humidity, atmospheric pressure, and vibration level.

**Temperature**. There are multiple failure mechanisms in electronic components related to their temperature: electromigration (*d'Heurle, 1971*), high temperature stress migration (*Aoyagi, 2005*), thermal fatigue (*Zhou & Hashida, 2002*), mechanical stresses induced by differential thermal expansion of materials (*Rabiei & Evans, 2000*), the drift of parameters
(frequency, current, voltage) of devices (*Rabiei & Evans, 2000*), solder joint failures (*Yeh et al., 2002*), ionic effects (*Berg & Paulson, 1980*), increase in leakage current (*Hamidi & Coquery, 1997*) and bond-wire fatigue (*Roesch & Jittinorasett, 2004*; *Matsunaga & Uegai, 2006*). The detailed review of electronics failure modes that are influenced by temperature is given in *Blanks (1990)*. The temperature of individual simulated components we consider as an internal factor and, therefore, not implement in the SANgo code. Inclusion of all or even some of these mechanisms into simulation would require a very detailed simulation of the circuit boards and electronic parts of the SAN components.

In general, the Arrhenius model (*Lakshminarayanan & Sriraam, 2014*) is a good approximation for exact failure mechanisms, including electromigration, corrosion, and certain manufacturing defects when dealing with slowly changing ambient temperature. The model is derived from the observed dependence of chemical-reaction rates on temperature changes. We use the Arrhenius model solely to take into account the external factor of ambient temperature for the failure rate of the electronic components. According to this model, the reaction acceleration rate is given by:

$$K = A\exp\{E_a/k[1/T_{ref} - 1/T]\}, \tag{1}$$

where $K$ is the resulting failure rate, $A$ is a rate constant empirically derived, $E_a$ is the activation energy ($eV$), $k$ is the Boltzmann's constant ($8.6 \times 10^{-5} eV/K$), $T$ is the ambient temperature (K) and $T_{ref}$ is the component's reference temperature (K).

For non-electronic components (such as hard-drives), the failure rate is calculated based on fail-safe operation time $T$ (*Sankar, Shaw & Vaid, 2011*). Using the maximum likelihood method, $\beta$ (form factor), $\zeta$ (scale coefficient) the failure probability is estimated:

$$K_{\beta,\zeta} = \frac{\beta T^{\beta-1} \exp{-\frac{T^\beta}{\zeta}}}{\zeta^\beta}, \tag{2}$$

where $\beta > 0, \zeta > 0, T > 0$.

*Humidity. Atmospheric pressure. Vibration.* The impacts of these three external factors are modeled similarly. The predefined mappings from the environment factor value to failure rates for humidity, pressure, and vibration are taken from *Mitchel (1996)*, *Strom et al. (2007)* and *Dutta & Barnard (2017)* correspondingly. At each timestamp $t$ of the simulation, current values of `humidity`, `pressure`, and `vibration` are taken. Then the corresponding failure rates are calculated using the said relationships.

For all of the external factors the failure-triggering pseudo-code looks like:

```
WHILE simulation is going:
    READ current values of temperature, humidity,
                          pressure, vibration;
    CALCULATE failure rates imposed by:
                    f1 ← temperature;
                    f2 ← humidity;
                    f3 ← pressure;
                    f4 ← vibration;
```

```
GENERATE a random number R from [0, 1) uniformly;
IF R < (f1 + f2 + f3 + f4): CREATE breakdown;
ELSE: nothing happens;
WAIT until next time step;
```

## ILLUSTRATIVE EXAMPLES

An example of the SANgo application for the simulation of a mid-range storage array with basic structure is described below. The simulated architecture was configured with parameters corresponding to the real SAN components parameters. Most underlying effects of the SAN functioning, such as the operating system and software logic, load distribution algorithms, and other more complex hardware details are not described by the simulator. In order to better approximate the behavior of the real-life system, SANgo was coupled with DeepController (DC)—an RL optimizer. The scheme of cooperation between SANgo and DC is shown in Fig. 3.

DC represents the RL and deep learning paradigm for controlling the simulator by tuning the effective parameters of its components. The simulation process is split into two phases.

| | |
|---|---|
| **Training phase** | : DC takes as input the real SAN data (load and metrics), and the initial effective parameters of SANgo. By varying the effective parameters, DC learns what needs to be done to obtain a simulator state, similar to the real SAN under the same load conditions. The output for this phase is a trained model with corresponding neural network (NN) weights. |
| **Control phase** | : during the actual simulation, DC takes as input the load and current effective parameters and simulated metrics. The corrections to the effective SANgo parameters are given to the output for every next simulation step. |

In this context, it is important to note that a special wrapper was implemented to create an OpenAI Gym environment (*Brockman et al., 2016*) using DC, SANgo, real SAN data, and initial effective parameters. Such an environment provides an advantageous opportunity to use multiple state-of-the-art RL-algorithm libraries, implemented in the OpenAI toolkit (*Dhariwal et al., 2017*; *Hill et al., 2018*; *Kolesnikov, 2018*). In particular, we used a DDPG model to train DC as one of the efficient approaches for Continuous Control problem (*Lillicrap et al., 2015*). Also, the obtained gym-environment could be used as a benchmark for other RL-algorithms, especially since there are not so many environments for digital twins (*Koch et al., 2019*).

The real SAN data were used to validate the behavior of the hybrid simulator. A sequence of load requests was generated on the real storage system prototype together with artificially induced failures of one of the components: a storage controller, a network interface, or a storage device. The labeled metrics of the healthy and broken systems were collected and used to train the hybrid simulator with the same workload and failure scenarios. In the end, the simulated metrics are compared against the real ones. An example of one of the metrics –the storage controller CPU load is shown in Fig. 4.

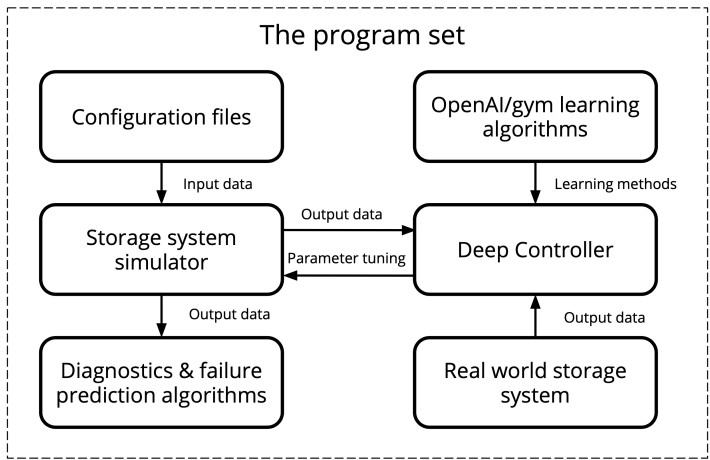

**Figure 3** The hybrid simulation scheme using OpenAI/gym interfaces.

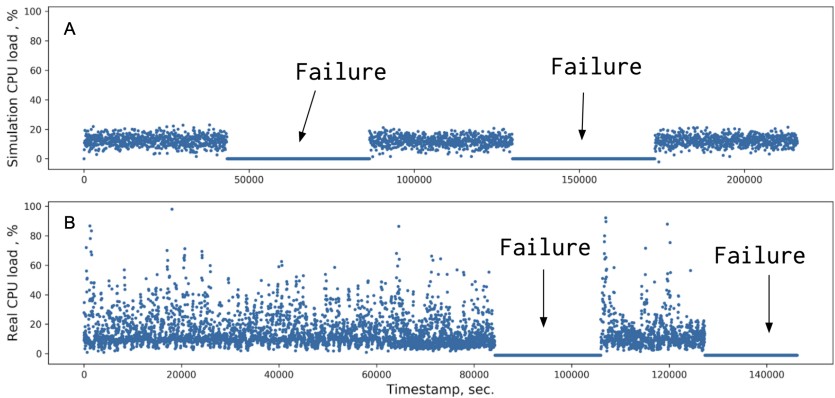

**Figure 4** Comparison of the CPU load metrics between simulated (A) and real data (B). The periods marked 'Failure' correspond to a storage processor being offline.

One can observe a qualitative agreement between the simulated and the real data. The more component parameters added, the closer the DC model distributions were to the real data.

For generating data in this scenario, the simulator was launched by the following command:

```
go run main.go -sim_run=210000 -platform=virt_setup.xml
        -num_jobs_config=num_jobs.json -packet=packet.json
        -controlling_mode=0 -num_jobs=1 -atm_dep=temp.json
        -atm_control=atm_control.json -file_amount_w=10
        -file_size_w=10GB -output=output.json
```

In this command snippet, several flags are used. They define the following behavior of the simulation. It runs for 210000 s, writes 10 files, each with size 10 GB, uses 1 job,

a topology information is provided in `virt_setup.xml`, the external dependencies are defined in `atm_control.json`, `packet.json` provides a packet block timing information, and finally, output data is written to the `output.json` file. There are other flags available to configure the simulator functionality. A complete list of execution options is given in Table 4.

The hardware parameters configuration is an essential step in the SANgo setup. For example, we used empirically obtained parameters of the storage devices affecting the speed of writing and reading data using Flexible I/O tester. We performed data transfer in different modes, such as reading, writing, random reading, random writing, reading/writing, and random reading/writing. These modes were combined with different data block sizes—from 2 Kb to $2^6$ Gb to cover a full range of possible IO conditions. The FIO tool allows user to directly measure some parameters, such as latency, transmission, and processing time. Other parameters, such as rate and seek time, had to be calculated from the parameters that FIO allowed to measure. An example of hard drive parameter values for 8 Kb block size, measured with such an approach, is provided in Table 5.

## IMPACT

The SANgo simulator helps researchers to address several issues in the area of SAN architecture development. The primary purpose of the developed SAN digital twin (*Sapronov et al., 2018*), in which the SANgo is a critical component, is to provide realistic data for the SAN optimization. In conditions where the real data are scarce, for example, during the early stages of development, the synthetic data may help to improve the system's stability, study its behavior under different load regimes and observe how the system reacts to various failures of its components. Besides, synthetic data allows the development of diagnostic and prediction tools for system malfunction based on data-driven algorithms available in the Machine Learning domain (*Hushchyn, Sapronov & Ustyuzhanin, 2019*). The SAN architecture optimization and parameter tuning is another possible application of such a digital twin.

For example, the SANgo code was used to produce synthetic data for training diagnostic and predictive software. The software relies on data-driven algorithms to assess the state of the storage system, diagnose its malfunction, and calculate the probability of failures in the near future. Using SANgo helps to significantly reduce the amount of collected real SAN operation data, which is quite expensive in most cases.

The application of SANgo software is somewhat limited outside the intended scope. The simulator intentionally describes the storage infrastructure in low detail and, therefore, can be used only for deterministic and approximate emulation of the SAN operation. However, the general approach of combining the basic simulator with an RL agent is very scalable for other uses where one needs to simulate a complex apparatus or a system with many loosely controlled parameters, but with available real data collected during its operation.

The SANgo alone has limited precision in terms of simulation quality. The storage system model describes only high-level components and provides a very simplified description of their failure mechanics. However, the strong side of our approach is the adjustable

**Table 4  A complete description of the SANgo execution options.**

| Type | Flag's name | Type | Description |
|---|---|---|---|
| Common | -sim_run | float | Total simulation runtime |
| | -disk_amount | int | Amount of hard-drives |
| | -num_jobs | int | The analog of the fio numjobs |
| Input files | -platform | string | |
| | -packet | string | Data block's timing parameters |
| | -atm_control | string | A change of atmosphere parameters over time |
| | -atm_dep | string | Impact of atmosphere parameters on the system components |
| | -client | string | |
| | -num jobs config | string | Client's load & quota specification |
| Load | -file_amount_w | int | Number of files to be written |
| | -file_size_w | float | Write files size range |
| | -load_range_w | float | File write rate |
| | -file_amount_r | int | Number of files to be read |
| | -file_size_r | float | Read files size range |
| | -load_range_r | float | File read rate |
| Anomaly | -anomaly_type | string | Controller, link, disk type anomaly |
| | -anomaly_amount | int | Number of anomalies |
| | -anomaly_tim_range | float | Anomaly ocurring rate |
| | -anomaly_duration | float | Duration of anomalies |
| DC[1] mode | -controlling_mode | bool | Enabling DC |
| | -host | string | Hostname |
| | -port | int | A communication endpoint |
| | -protocol | tcp, udp | SANgo-DC communication protocol |
| | -delay | float | Delay in messaging between GT and DC |

**Table 5  An example of hard drive parameter values for 8 Kb block size, where "t" is time in seconds. .**

| transmission t | latency t | read processing t | write processing t | sequential read rate t |
|---|---|---|---|---|
| 3.96e−05 | 0.01784 | 0.01229 | 0.02331 | 0.00065 |

parameters of these components. When the parameters are controlled by an RL-trained agent, the simulation quality can be improved. At the same time, the physical model within the SANgo framework ensures the results are interpretable and physically consistent.

While the software itself is not used in any commercial setting, the product of its application, the diagnostic and prediction tool for the SAN malfunction, is bundled with an undisclosed commercial storage system. The tool is developed and trained using real data and synthetic data obtained from the digital twin of the SAN.

## CONCLUSION

A flexible framework SANgo for the creation of an event-driven storage simulator was presented. Within this framework, the storage system models have simplified design, but researchers are allowed to adjust the components' effective parameters during runtime to improve the quality of the simulation. This approach complies with a hybrid simulation method, where a reinforcement learning algorithm adjusts a physical model.

The SANgo framework allows the user to emulate the I/O load on the storage system, as well as failures of its components. It was initially designed to work in tandem with the DeepController program, implementing the RL algorithm. The simulator can also be used as a benchmark for comparison of different learning algorithms, due to support of the OpenAI/gym interface.

The purpose of such a hybrid simulator is to serve as a storage system digital twin and provide large amounts of synthetic data. This data can be further used for storage system optimization, diagnostic, and failure prediction.

### Funding

The research was carried out with the financial support of the Ministry of Science and Higher Education of Russian Federation within the framework of the Federal Target Program Research and Development in Priority Areas of the Development of the Scientific and Technological Complex of Russia for 2014-2020 (unique identifier RFMEFI58117X0023, agreement 14.581.21.0023 on 03.10.2017). The funders had no role in study design, data collection and analysis, decision to publish, or preparation of the manuscript.

### Grant Disclosures

The following grant information was disclosed by the authors:
Ministry of Science and Higher Education of Russian Federation within the framework of the Federal Target Program Research.
Development in Priority Areas of the Development of the Scientific and Technological Complex of Russia for 2014-2020 (unique identifier RFMEFI58117X0023, agreement 14.581.21.0023 on 03.10.2017).

### Competing Interests

Ivan Tchoub and Artem Ikoev are employees of YADRO Inc., Russia. The authors declare there are no competing interests.

### Author Contributions

- Kenenbek Arzymatov conceived and designed the experiments, performed the experiments, performed the computation work, prepared figures and/or tables, and approved the final draft.
- Andrey Sapronov conceived and designed the experiments, authored or reviewed drafts of the paper, and approved the final draft.

- Vladislav Belavin performed the experiments, performed the computation work, prepared figures and/or tables, and approved the final draft.
- Leonid Gremyachikh performed the experiments, prepared figures and/or tables, and approved the final draft.
- Maksim Karpov analyzed the data, prepared figures and/or tables, and approved the final draft.
- Andrey Ustyuzhanin analyzed the data, authored or reviewed drafts of the paper, and approved the final draft.
- Ivan Tchoub and Artem Ikoev conceived and designed the experiments, authored or reviewed drafts of the paper, and approved the final draft.

### Data Availability
SANgo, a discrete-event based simulator, is available at Code Ocean: https://codeocean.com/capsule/9707185.

Code is available at GitHub: https://github.com/HSE-LAMBDA/sango.

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
