# Peer review of "SANgo: a storage infrastructure simulator with reinforcement learning support"

_PeerJ Computer Science, doi:10.7717/peerj-cs.271_

## Round 0.1 · original submission · Major Revisions

Based on the reviewers’ comments and my own evaluation, the authors are requested to submit a revised version of the manuscript by especially focusing on the following issues (in addition to the individual comments of the reviewers).

(1) Please check the spelling and grammar issues.

(2) Please check the availability of the used software.

(3) Please give more detailed insights into the advantages and possible shortcomings of SANgo.

·

Basic reporting

Numbered points are in order of precedence, bullets signify points to be
addressed in no order of precedence.

Minor Spelling and Grammar issues (indicative, non-exhaustive):

- 22: Replicated effects should be clarified
- 62: "method of hybrid simulation" -> "The hybrid simulation technique"
- 101: Replace (Fishman) with the reference
- 107: "attributed metric information" -> "extra metadata describing attributes"
- 133: Reference missing (can be found in ?)

Experimental design

1. The data required for the illustrative example needs to be provided, along with details of how the data was obtained.

Validity of the findings

Numbered points are in order of precedence, bullets signify points to be
addressed in no order of precedence.

1. 145: The Arrhenius model is not justified in this context. The reference
is not of a journal either, and also does not account for the variation in
the chemicals and junctions used in computer circuits.
2. 155: Similar concerns exist, however modeling these effects with an
exponential is more reasonable due to the very limited variation of these
factors in production.

The authors are directed to the following:

- https://ieeexplore.ieee.org/abstract/document/62568
- https://ieeexplore.ieee.org/abstract/document/275221
- https://ieeexplore.ieee.org/abstract/document/4292056
- https://www.springer.com/gp/book/9780387257624

Furthermore, there are several concerns with modeling temperature in this
manner. The nature of the Arrhenius equation, as well as the operating
temperature loads of most IC devices do not correlate well. For the Arrhenius to
be justified, the correlation between the activation energies of the
semiconductor chips used in the datacenter model correspond to the operating
temperatures.

Additional comments

The software URL (https://gitlab.com/lambda-hse/sango) is not publicly
accessible. This makes the software more complicated to test (via the codeocean
capsule). This needs to be rectified, especially since the codeocean capsule
lists the software under the permissive GNU-GPL_v3. Furthermore, the gitlab URL being inaccessible has the unintended effect of invalidating the installation
instructions. Though not necessary for this publication (nor has it factored in
my recommendation), a good API documentation should be ideally present. The
`bib` file should have been sanitized before usage, possibly by filtering via
Zotero, Mendeley, Jabref or others. More details of the data used is required.
The specific usage of OpenAI models should also be clarified.

The code is well written, and the work presented is worthy of inclusion.
However, given the severe misgivings regarding the modeling of environmental
effects, in its current form this is not ready for publication.

Reviewer 2 ·

Basic reporting

The authors should demonstrate an adequate understanding of the relevant literature in the field and cite an appropriate range of literature sources.
The paper contains significant information adequate to justify publication.

Experimental design

no comment

Validity of the findings

no comment

Additional comments

The authors should check again the paper. For example, on page 8 the number of the table that shows the full flag list is not mentioned. The conclusion should be checked too.

Annotated reviews are not available for download in order to protect the identity of reviewers who chose to remain anonymous.

---

## Round 0.2 · accepted · Accept

I am happy that the authors have revised the manuscript according to the reviewers' and my comments. I think it is ready to accept the manuscript.

·

Basic reporting

No comment

Experimental design

No comment

Validity of the findings

No comment

Additional comments

I thank the authors for the improvements to their initial manuscript.

Reviewer 2 ·

Basic reporting

No comment

Experimental design

No comment

Validity of the findings

No comment

Additional comments

The paper contains significant information adequat to justify its publication